# High Penetration of Renewable Energy Sources and Power Market Formation for Countries in Energy Transition: Assessment via Price Analysis and Energy Forecasting

**Dimitrina Koeva [1,*], Ralena Kutkarska [2] and Vladimir Zinoviev [3]** (ID)

1   Department of Electric Power Supply and Electrical Equipment, Technical University of Gabrovo, 4 Hadgi Dimitar Str., 5300 Gabrovo, Bulgaria
2   IT Department, Municipality of Sliven, 1 Tsar Osvoboditel Blvd., 8800 Sliven, Bulgaria; ralena.dimitrova@gmail.com
3   Department of Economics of Infrastructure, University of National and World Economy, 19 8-th December Str., 1700 Sofia, Bulgaria; vzinoviev@unwe.bg
*   Correspondence: dimitrina.koeva@gmail.com

**Abstract:** Climate change as a challenge we all are facing, varying degree of economic development as a result of COVID-19, the volatility in energy prices and political as well as other factors, most countries have restructured their electricity markets in order to facilitate the use of green renewable energy. The right energy mix in a period of energy transformation is the best strategy for achieving reduction of carbon emissions. Bulgaria is a special case because it has expanded the use of solar and wind energy exponentially, without conducting an adequate preliminary forecast analysis and formulating a parallel strategy for the development and expansion of the energy storage infrastructure. In this regard, the article is focused on how the power energy market is structured with the increasingly large-scale and global penetration of renewable energy sources as primary energy sources, observing several key factors influencing the energy transition. Due to the cyclical nature of energy production and the necessity for a smooth and efficient transition, a long-term seasonal storage plan should be considered. Furthermore, solar energy production facilities have a greater share of installed power, but wind power facilities generate a roughly equivalent amount of electric energy over the course of a year. One of the aims of this research is to discover an appropriate model for predicting the electricity output of wind and solar facilities located in Bulgaria that can be used to ease the transition process. Based on thorough data analysis of energy production over the past 11 years and 5 months, our findings suggest that a SARIMA model might be appropriate, as it takes into account the seasonal cycles in the production process.

**Keywords:** energy transition; renewable energy; energy forecast

## 1. Introduction

The use of renewable energy sources and the prudent consumption of energy are crucial for the sustainable development of any economy: they ensure the achievement of energy supply security goals, reduce the dependence on sudden changes connected with oil prices, contribute to reducing the trade imbalance and stimulate the creation of new jobs.

Renewable energy sources play a key role in the ongoing green energy transition. They are also a way for EU countries to become less dependent on imported energy resources and on energy market disruptions that cause an increase in prices. This is how the geopolitical energy battle has inevitably led to an increased demand for clean energy, as we are moving away from the dependence on fossil fuels and the dependence on Russian imported fuel.

The electricity market for countries that are at the beginning of their energy transition (including Bulgaria) is in the process of gradual liberalization. One of the challenges that Bulgaria is facing is in the low-carbon economy area. The green transition of the country is

set out in the National Recovery and Resilience Plan of Bulgaria, which is currently being implemented. It has been indicated that the energy sector is the biggest source of greenhouse gas emissions on a national level, with over 70% of the total emissions of the country. Coal-fired thermal power plants are responsible for almost half of the sector's emissions. The effort to decarbonize the economy will require a comprehensive reform of the national energy sector, but it will also imply a significant need for investment [1]. Adequate technical means for energy storage will be needed in order to make any progress and achieve long-term sustainability in the energy transition, combined with low electricity prices.

In order for the energy transition to be successful, factors such as climate changes, innovation in technology, business processes, policies and the current levels of economic growth in each country must be taken into account [1–7]. With this fact in mind, the influence on the energy intensity of the industry as an important factor for carbon neutrality has been investigated in [8–10], and carbon targets can be met by using RES. The activities and policies that have to be initiated to manage the process of switching from fossil fuels to RES as sources of electricity from April 2009 to July 2021 are presented by introducing the so-called RES deployment rate index [7,11]. There is no detailed study regarding the energy reforms that are needed for the transition to clean energy in China [11], nor is there one for Bulgaria. Several key factors are mentioned: the introduction of "dual carbon goals" (peak carbon savings by 2030 and carbon neutrality by 2050); regional and national policies to promote green energy consumption; and developing the spot-, mid-, and long-term markets for green electricity. Since there is no properly structured and balanced market without predictive analysis, our analysis and research can be structured into the following main sections to meet the aims of the article: 1. Renewable energy growth and carbon emissions: a global and regional view; 2. Models and approaches for building a forecast analysis of renewable energy generation in Bulgaria; 3. Analysis and discussion of the results in terms of adequacy, accuracy and application of the selected model.

## 2. Technical and Economic Considerations—General Statement and Approach

Electricity costs depend more on the cost of energy storage capacity than on the capacity of power plants. Meeting the demand for electricity with other alternative sources during 5% of the hours can reduce electricity costs by half. Solar and wind power can help decarbonize the electricity generation process, but in order to meet this demand, energy storage technologies are also needed. There are systems that combine intermittent RES with storage and other technologies, and when comparing their electricity costs to alternative solutions, it is estimated that in regions with high resource and optimal resource combinations, low-cost energy storage capacity (<20 USD/kWh) is needed for cost-effective and reliable generation of baseload electricity. However, when other technologies meet the demand limit of 5% and even with significantly more expensive storage, costs could still be cut in half.

The advantages of RES during an energy transition are clear: greater energy independence, a more predictable and sustainable electricity market, lower prices and overall increased security for businesses and households. Given that wind and solar projects are expected to be the backbone of the transition to renewables in Europe and considering global decarbonization, these two energy sources are guaranteed to have a sharp rise. The start of this trend is noted in the International Energy Agency (IEA) report: the investment in low-carbon energy technologies had to be approximately 500 billion USD/year by 2020 and then it will double again to 1 trillion USD by 2030 [2].

In order for Europe to meet its targets for the 1.5 °C scenario (Figure 1), a major transformation of the design of the electricity market will be needed. In addition, there are also some financial implications—investments of 5.7 trillion USD/year by 2030. Since the investment decisions are long-term and the stranded asset risks are high, decisions must be guided by long-term logic—the cumulative investment in the energy transition will need to reach more than 115 trillion USD by 2050 [3,4]. The Global Energy Transition Outlook estimates that 0.7 trillion USD in annual fossil fuel investments should be redirected

to energy transition technologies. Although a large amount of the additional capital is expected to come from the private sector, doubling of the public funding will also be needed in order to speed up the private financing process and create an enabling environment for the transition that delivers optimal socio-economic benefits.

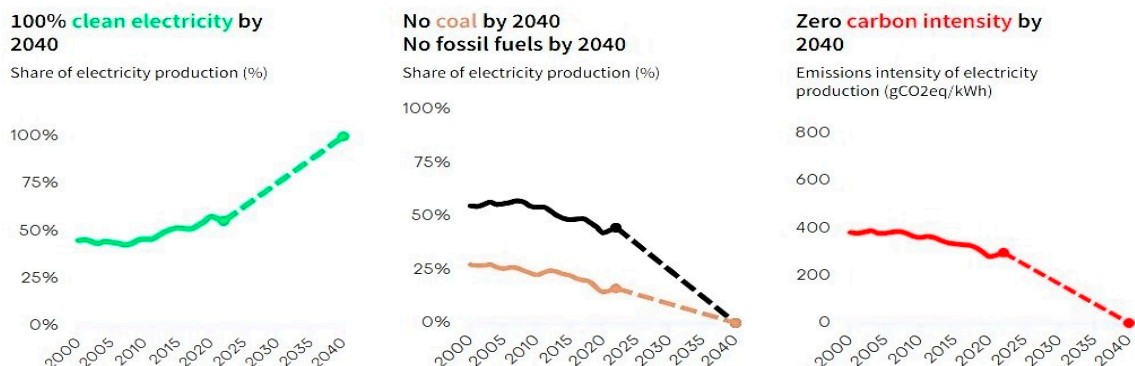

**Figure 1.** European progress towards 1.5 C benchmarks in the energy sector, 2000–2040 [3].

The period from 2022 to 2027 is a period of energy transition for many countries, and according to the IEA forecasts, around 1500 GW of solar and over 570 GW of the new onshore wind and power capacities will need to be commissioned. Wind and solar renewable sources will therefore be responsible for 20% of global electricity generation. The slow growth of wind power plants is due to the long permit-issuance process and the lack of improvements in the transmission grid infrastructure. Globally, the installed capacity and the number of offshore wind farms is increasing, but in Europe, their share will decline from 50% in 2021 to 30% by 2027 [5].

This phenomenon can be explained by the many challenges related to policy, regulations and funding. Political and regulatory uncertainty remains the main obstacle for countries that are currently embarking on their energy transitions. The forecast of the IEA analysts is that the annual renewable capacity additions after 2022 will range between 350 and 400 GW, as wind and solar energy will be 85–90% of all new installed capacity.

Another important issue: the rapid expansion of the wind and solar industries over the next 5 years will be heavily dependent on the module prices for their structures, which are currently 25–30% higher compared to 2020. Prices of the so-called "critical materials" [12] (cobalt, copper, nickel, lithium, rare earths, neodymium) are rising (Table 1), but since their market is unstable, strategies to mitigate the dependencies on critical materials will be needed.

**Table 1.** Expected growth in the production of critical materials.

| Material | 2020 [Mt/Year] | 2050 [Mt/Year] |
|---|---|---|
| Copper | 30 (8.5 recycling, 21.5 primary production) | 50–70 |
| Nickel | 2.54 | 5–8 |
| Lithium | 0.41 | 2–4 |
| Cobalt | 0.14 | 0.5–0.6 |
| Neodymium | 0.03 | 0.2–0.5 |

Trends in the annual share of RES capacities can be seen in Figure 2 [5,13]. The general conclusion that can be drawn is that the growth of RES capacities in Europe is constrained by three main challenges: inadequate support/financing schemes, long and complex permitting procedures and slow pace of transmission and distribution of grid modernization. Then how is Europe supposed to meet the REPowerEU targets? According to [5,13], in order to meet these targets by 2030, 592 GW of solar and 510 GW of renewables are needed, which is 69% of the overall share of renewables. On an annual basis, an increase of 48 GW of solar and 36 GW of wind energy is required. For the period 2022–2027, the

estimated average growth of the net capacity is 39 GW of solar energy and 17 GW of wind
energy, for a 54% share of RE capacity. Therefore, we observe a lag of 15% (Figure 3).

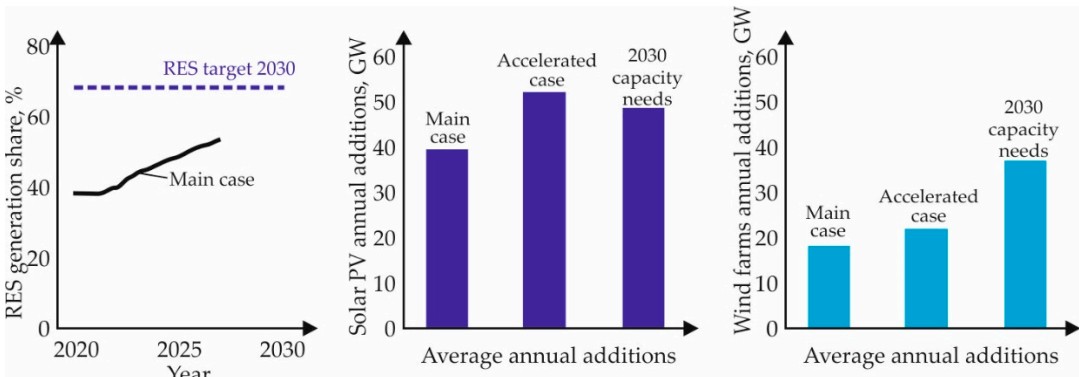

**Figure 2.** Renewable electricity shares in the main case; average annual addition for solar and wind,
2022–2027 [12].

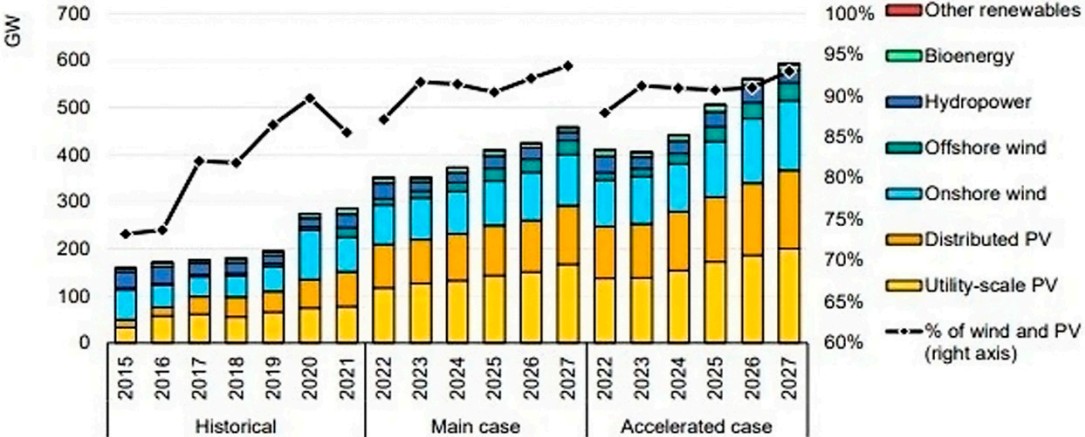

**Figure 3.** RES generation annual net capacity by technology, main and accelerated cases,
2015–2027 [5,13].

Another important aspect that needs to be taken into consideration if we want to make
progress in energy transition is the careful analysis of the trends in renewable energy costs
and prices. Here, there are problems as well: insufficient and publicly unavailable large
amounts of important data, and differences in approaches and methodologies in modeling
the interdependent variables and factors that have an influence on the generated power.

In Europe, the rise in fuel prices and the rise in $CO_2$ allowance prices in the European
Emissions Trading Scheme (ETS) have led to a sharp increase in the cost of electricity
generation. These prices have increased from 40 USD/t in early 2021 to 99–114 USD/t in
the first 45 days of 2022 [14,15].

In the first 45 days of 2022, only the fuel cost (along with $CO_2$ permits) of a coal-fired
power plant increased by about 61–92 USD/MWh compared to the previous year. The total
fuel cost of a coal-fired power plant in Europe is between 117 and 148 USD/MWh. The in-
crease alone is higher than the total life cycle cost of electricity for new onshore RES projects
in Europe. When it comes to natural gas, the climb is even more dramatic, with the fuel cost
of electricity produced by a combined cycle gas plant increasing by 133–167 USD/MWh
from early 2021 to a total of 186–220 USD/MWh. Even if prices decrease, the negative eco-
nomic effects caused by periods of high prices are still a fact. Renewable energy generation
costs are lower than coal plants, even taking into consideration the cost of technical storage.
The BloombergNEF (BNEF) battery price index fell by 89% between 2010 and 2021: from
1200 to 132 USD/kWh [16]. The price decline can be seen in Figure 4.

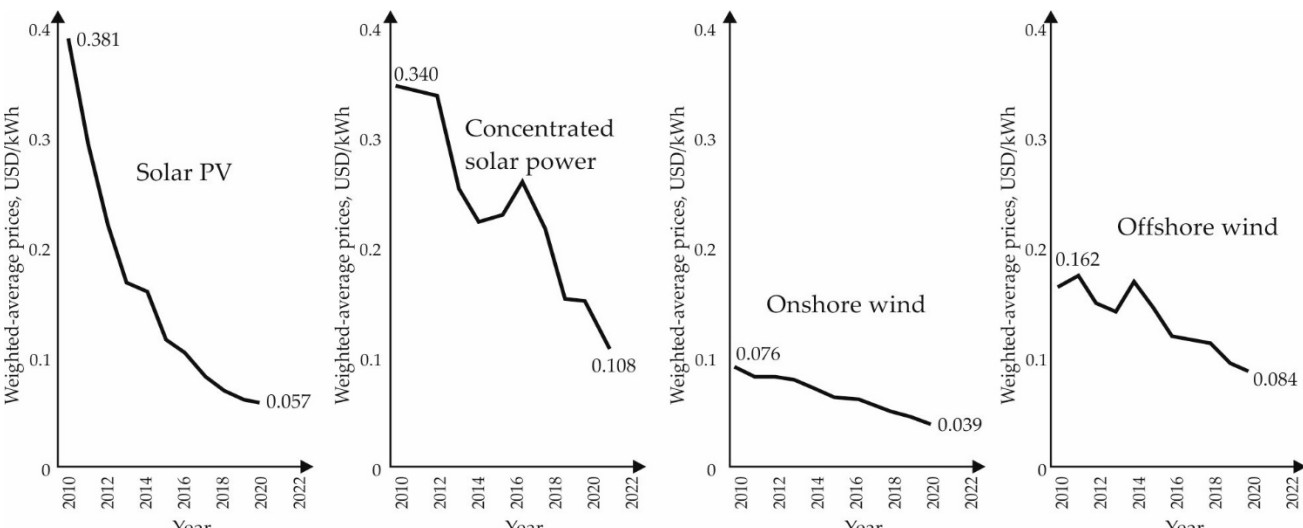

**Figure 4.** The global weighted-average prices for solar, concentrated solar, onshore wind and offshore wind power, 2010–2023, [16].

Achieving an energy transition in line with the 1.5 °C scenario will require cumulative investments of around 131 trillion USD between 2021 and 2050. In the short term, it is estimated that investments will have to reach 5.7 trillion USD/year between 2021 and 2030, including a shift of 0.7 trillion USD/year from fossil fuels to energy transition technologies. Between 2031 and 2050, an average of about 3.7 trillion USD/year will be needed. Table 2 provides a breakdown of the annual investment needed in the short term (2021–2030) and in the long term (2031–2050) by technology pathway [3].

**Table 2.** Required investments for technological equipment on an annual basis (CCS—carbon capture and storage; BECCS—biomass coupled with CCS) [3].

| Technological Avenue | Investment Needs, (Billion/Year) | |
|---|---|---|
| | 2021–2030 | 2021–2030 |
| RES capacity | 1045 | 897 |
| Direct use of RES, including heat | 284 | 115 |
| Power grids and energy flexibility | 648 | 775 |
| Energy efficiency (including industry): | 2285 | 1106 |
| – charging station infrastructure for EVs | 86 | 153 |
| – heat pumps | 154 | 77 |
| Electrification in end-use sectors | 240 | 229 |
| CCS and BECCS | 41 | 77 |
| Fossil fuel, nuclear, innovation, etc. | 1010 | 321 |

At the beginning of its energy transition, Bulgaria had an energy mix with an increasing share of wind and solar energy (Figure 5). For this reason, technologically and technically adequate energy storage is needed to facilitate this transition to systems whose sources are difficult to predict and have an intermittent nature. For our country, energy storage appears to be a key factor for the development of peaking capacity, balancing, energy displacement, frequency regulation and system services for the electricity transmission of the infrastructure.

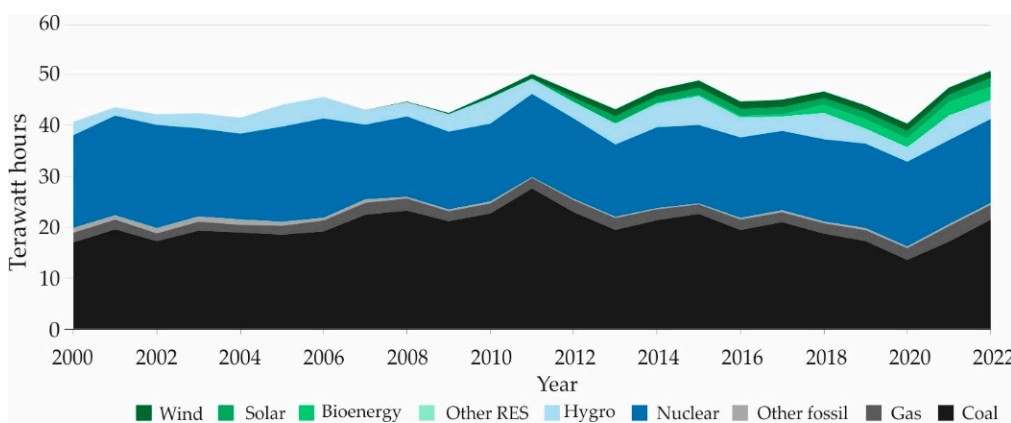

**Figure 5.** Bulgarian electricity generation by source, TWh (Ember Electricity Data Explorer, www.ember-climate.org) (accessed on 3 September 2023) [3].

Wind and solar power overtook gas as a main energy source for the EU for the first time last year, reaching more than 1/5th of the electricity generated in 2022, according to a report by energy experts [17] (Figure 5). Are we close to the time when fossil fuels can finally be fully replaced by renewables? The war in Ukraine has fundamentally changed the energy landscape in Europe, particularly increasing and accelerating the transition to renewable energy sources (RES) and minimizing dependence on Russian gas. Thus, the latest European Electricity Review found that a record 22% of the EU consumption was generated only by wind (15%) and sun (7%) in 2022 and roughly the same was the share of nuclear generation, whereas the share of fossil gas has fallen to 20%. For 2023, it is estimated that the growth of solar and wind energy will continue, while the generation of fossil gas will decrease by about 20%.

Since March 2023, there has been a new policy framework for EU countries in the context of electricity storage and grid flexibility. The requirements for all members (including Bulgaria) are as follows: to assess the flexibility of electricity grids in order to make the energy transition possible; to set guiding national targets for energy storage in order to adequately finance it; and to set up and implement schemes to support system flexibility services (Greece and Hungary are already implementing such measures). Hybrid renewable energy auctions, which are successfully implemented in Germany, together with storage systems have proven to be a practical approach to mitigate the risks associated with the accelerated entering of renewables in the grid: forced shutdowns and limitations from green power generation have been reduced and the dispatch of renewables and the adequate system services have been improved.

In the past two years, Bulgaria doubled its installed solar capacity to 2.2 GW and by the end of 2030, another 700 MW is expected to be commissioned. The implemented REPowerEU plan provides electricity from renewable energy at the lowest prices and this is already a fact. However, the biggest challenge is the lack of initiatives and actual storage capacity. After this prolonged boom in solar energy investment (a 30% surge at the expense of a 58% year-on-year collapse in coal-fired electricity generation), its share in the energy mix has become so large that hours of intense sunshine have produced surpluses of energy and Bulgaria has witnessed the phenomenon of negative prices for the first time this year. Energy markets follow their natural logic and look for the most cost-competitive sources of generation—renewable energy. On 20 May 2023/13–16 h and on 21 May 2023/10–17 h, 1 MW cost 0.00 BGN (exchange quotes), which was due to low demand and overproduction.

Market dynamics, of course, favor more economically competitive technologies. Renewable energy, through its continuously decreasing capital and operating costs, provides the lowest electricity prices, but its integration into the electricity system creates problems and additional challenges in terms of flexibility and proper dispatch, as there is no adequate storage capacity built. For 2022, according to the BG Energy System Operator (ESO), the

total installed capacity of generating systems installed is 13,505 MW, with a total electricity generation of 50,578,798 MWh [18,19]. A more detailed presentation of the data is shown in Table 3.

**Table 3.** Generating capacities and gross electricity generation for 2022.

| Power Type | Installed Capacity, MW/Percentage of Total, % | Gross Electricity Generation, MWh | Change for 2022/2021, % |
|---|---|---|---|
| Nuclear power plant | 2000/14.8 | 16,464,662 | 0 |
| Coal plants | 4475/33.1 | 26,463,339 | 0 |
| Water plants | 3214/23.8 | 3,810,674 | 0 |
| Wind power plants | 705/5.2 | 1,499,125 | 0 |
| PV power plants | 1726/12.8 | 2,022,607 | 38.5 |
| Biomass power plants | 77/0.6 | 318,391 | −1.5 |

IRENA and NREL use some indicators to assess the potential of RES in Bulgaria, Figure 6, [20,21]. Bulgaria strives to pursue a coherent European energy policy in line with its climatic, geographical and economic realities. One of Bulgaria's most urgent targets, aimed at a green energy transition, is the decarbonization plan, according to which the average emissions from our energy production should fall to below 350 kg/MWh by 2030. Currently, our coal plants produce emissions of over 1100 kg/MWh, but through a mix of different technologies, the transition can happen smoothly and without creating risks for the country's energy system.

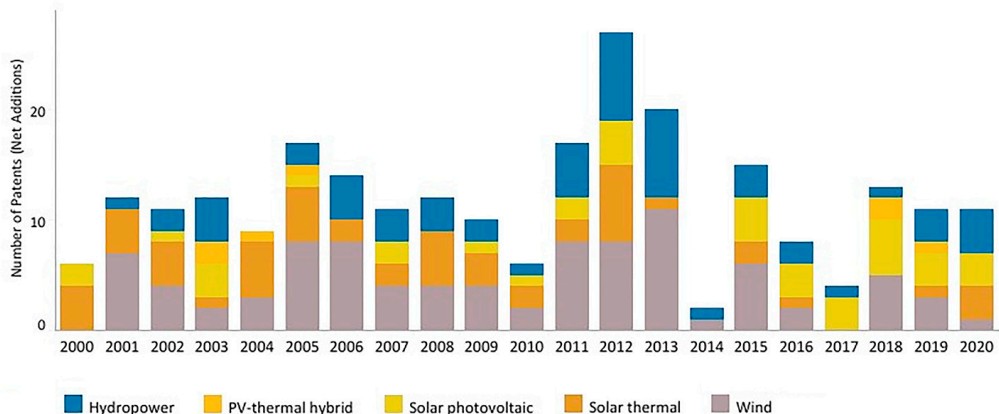

**Figure 6.** Bulgaria's renewable energy patterns evolution (www.irena.org, last updated 1 November 2022).

For constructive and technological reasons, wind (WPSs) and solar (PVPSs) power stations worsen the power quality when it comes to grid connection by causing voltage fluctuations, increased harmonics, the appearance of flicker (jitter), and dynamic variation of the flow distribution. They also have a negative effect on system stability during disturbances and post-emergency modes. At the same time, WPSs and PVPSs cause frequent variation in the power output of the operating units of thermal power plants, which disturbs the normal operation of the equipment, and treatment plants and can have a reverse environmental effect.

For the same reasons mentioned above, RES cannot participate in primary and secondary frequency regulation and cannot be relied upon for emergency management of electrical energy systems (EES) and restoration of EES after severe accidents.

In terms of real-time management of the EES, without disrupting the schedules for inter-system exchanges with neighboring ENTSO-E countries (the European Network of Transmission System Operators), the ability of our power system to connect WPPs and PVPSs is limited and is determined by the currently available regulation capacity and the

available regulation range. However, in the present circumstances, investors with contracts and paid guarantees significantly exceed this capacity.

The connection of RES to the EES is a problem not only in the Bulgarian EES but also in all other countries. RES are usually built in areas where there is no transmission grid or the existing grid is sized to supply small electrical loads. Under existing regulations, due to the slow procedure for purchasing and changing the use of the necessary land, it is not possible to build new power lines and substations at the pace of RES construction. At the same time, it is difficult to reconstruct and develop the existing electricity transmission network before the necessary new power lines are constructed because the security of the electricity supply is reduced and the risk of accidents in large areas of the country increases. With this fact in mind, land acquisition is considered a priority in the construction of transmission lines and substations for the development of the RES sector.

Despite the problems and limitations, Bulgaria maintains an "energy dialogue" with the neighboring countries. RES also play a role in this energy exchange, as shown in Figure 7. The carbon intensity of exports and the share of RES in the energy mix are also presented.

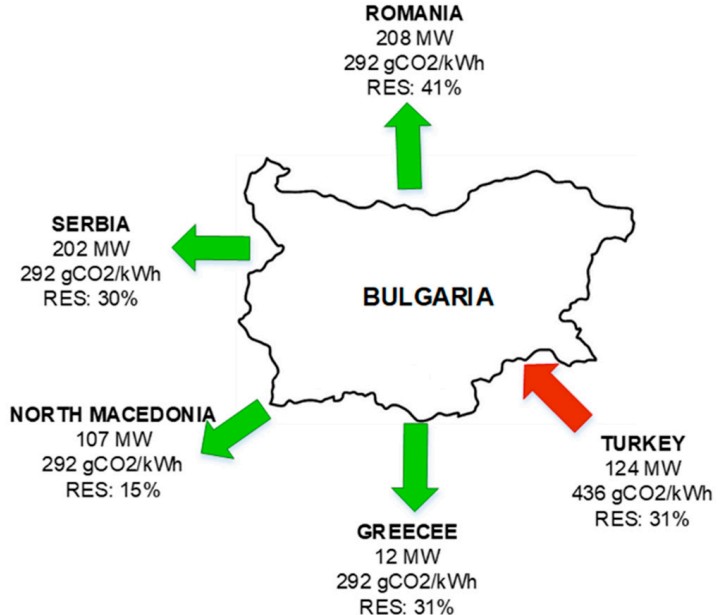

**Figure 7.** Energy exchange of Bulgaria for 24 August 2023/15:00 h [18].

The ESO presents up-to-date data with a dynamic label where the energy exchanges with bordering countries, 24 h consumption load charts, and energy price trends can be seen (Figures 8–10). The composition of the energy mix of the country is different for each day and time (Figure 8c). The same goes for energy exchange with the neighboring countries. For the observed day, the energy mix includes nuclear energy as a baseline source and coal plants as a secondary baseline source, which is responsible for balancing the energy and sustainability of the EES. RES are presented by solar and wind energy during the observed day. The pumping storage (hydro) plants take a peak load between 9:00–11:00 and 18:00–22:00. On this day, there was electricity export mostly to Romania and Serbia.

Another very important point for Bulgaria is to use its own energy sources without exploiting the supply of such sources because this leads to dependencies, which in some cases could be a problem. The basic concept is to do this with a minimum usage of natural gas, which during the transition period can be replaced by Bulgarian fossil fuels (such as lignite) and the quantity used should be gradually reduced over time.

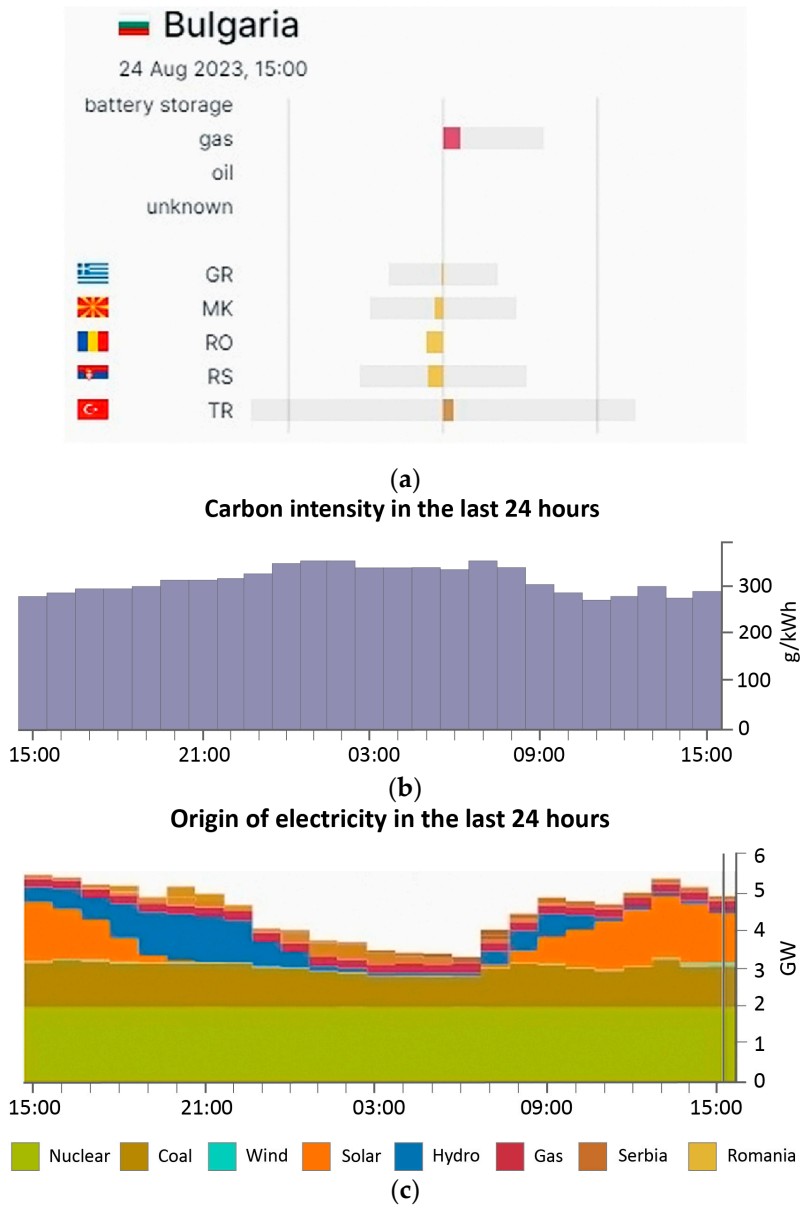

**Figure 8.** (**a**) Bulgaria's energy profile, 24 August 2023; (**b**) Carbon intensity in the last 24 h; (**c**) Origin of electricity in the last 24 h [18,22].

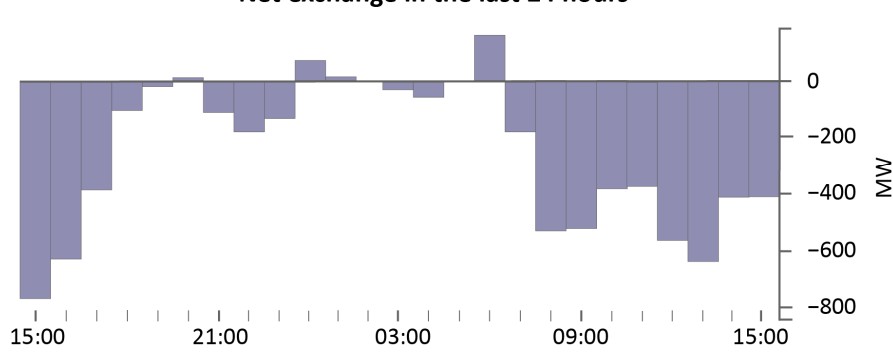

**Figure 9.** Bulgaria's net exchange in the last 24 h, 24 August 2023 [MW] [18,22].

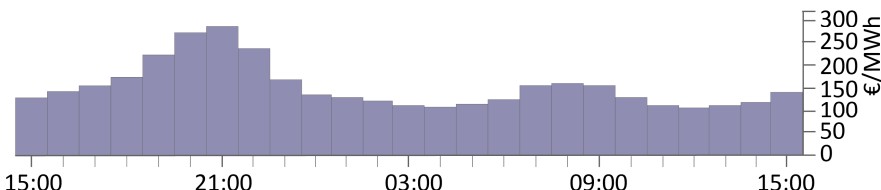

**Figure 10.** Bulgaria's electricity prices in the last 24 h, 24 August 2023 [€/MWh] [18,22].

The problem with renewable energy sources (RES) is that they are intermittent, which brings in the definitional requirement of efficient energy storage. Bulgaria has advantages in this regard, and one of them is the largest pumped storage hydroelectric power plant (pumped storage) in the region of the Chaira Dam. It is projected to be significantly expanded in order to include another reservoir.

Other options include compressed air energy storage and, of course, battery storage. Bulgaria is one of the largest producers of lead–acid batteries. This type of battery allows over 98% of the battery to be recycled, whereas for lithium-ion batteries there is still no such technology introduced.

There is also a great possibility for the huge "Maritsa" coal complex to transform and accept green energy production. The area there is extremely favorable, and the sunny location is perfect for green energy generation through the deployment of photovoltaics. Another advantage is the possibility of using the infrastructure that already exists for the production and storage of green hydrogen, too.

Bulgaria has the potential to develop all existing technologies for energy generation and storage, as well as to develop new ones. The necessary resources for this are available, as is the interest in investments from business alongside public funds.

With more support from the government, Bulgaria could become independent from the global energy markets in ten years, with renewable energy sources dominating and massive storage providing balance and the ability to export electricity. With cheap green energy, the opportunities for industrial and economic development are quite different.

From all that has been said so far, the imperative importance of creating accurate models for forecasting the amount of electricity generated from RES in the period of energy transition becomes clear. This is a key factor for building and operating smart grids and for accurately assessing the degree of the power-transmission system load. Last but not least, it is crucial for selecting appropriate electricity storage systems.

### 3. Materials and Model Description

This research is based on data about electrical energy produced by solar panels and wind generators in Bulgaria. According to the Sustainable Energy Development Agency—SEDA (executive agency within the Ministry of Energy of Bulgaria) [23], as of 2023, the number of solar renewable sites is 7719, having total installed power of 2740.3 MW and 193 renewable wind sites with total installed power of 706.4 MW. Together, they comprise 96.21% of the facilities currently used to generate renewable electricity in the country.

The data comprise 137 monthly observations covering a period of 11 years. Each observation contains the sum of the electrical energy produced for the given month. The values for the solar sources are provided in Table 4. The values for the wind sources are provided in Table 5. Both are graphically represented in Figure 11. The dataset is publicly available on the portal for electronic administrative services of SEDA [23].

**Table 4.** Energy produced (GWh) by solar sources.

| YY/MM | Jan | Feb | Mar | Apr | May | Jun | Jul | Aug | Sep | Oct | Nov | Dec |
|---|---|---|---|---|---|---|---|---|---|---|---|---|
| 2012 | 9.152 | 10.186 | 25.475 | 32.200 | 44.244 | 81.146 | 127.730 | 136.110 | 113.537 | 99.254 | 53.904 | 45.756 |
| 2013 | 53.168 | 53.902 | 103.883 | 126.280 | 165.495 | 153.386 | 172.085 | 169.971 | 143.858 | 115.095 | 63.432 | 71.433 |
| 2014 | 53.269 | 75.439 | 98.737 | 107.449 | 149.636 | 148.256 | 163.868 | 163.399 | 119.716 | 91.064 | 41.999 | 43.817 |
| 2015 | 62.901 | 76.527 | 96.862 | 145.820 | 153.814 | 150.008 | 179.143 | 158.420 | 121.292 | 81.297 | 83.623 | 71.953 |
| 2016 | 65.403 | 78.438 | 106.968 | 144.863 | 142.381 | 157.604 | 174.040 | 157.707 | 132.175 | 88.708 | 68.690 | 71.012 |
| 2017 | 42.502 | 88.145 | 118.285 | 144.630 | 151.824 | 157.233 | 163.611 | 165.707 | 131.857 | 124.032 | 53.800 | 61.345 |
| 2018 | 70.132 | 51.918 | 91.031 | 152.397 | 162.560 | 144.399 | 152.303 | 170.361 | 138.418 | 112.373 | 47.454 | 49.442 |
| 2019 | 54.167 | 83.649 | 138.841 | 130.446 | 149.856 | 156.620 | 164.999 | 171.264 | 138.484 | 122.038 | 49.581 | 57.341 |
| 2020 | 84.769 | 94.319 | 114.752 | 154.802 | 150.313 | 154.757 | 176.475 | 167.853 | 150.893 | 109.549 | 76.068 | 34.168 |
| 2021 | 61.393 | 89.997 | 120.437 | 142.536 | 170.985 | 152.485 | 189.521 | 182.622 | 138.306 | 96.752 | 69.400 | 53.013 |
| 2022 | 88.440 | 106.300 | 149.823 | 175.073 | 215.936 | 196.592 | 233.699 | 201.029 | 202.427 | 184.508 | 92.116 | 72.992 |
| 2023 | 69.465 | 139.207 | 180.321 | 196.610 | 214.919 | - | - | - | - | - | - | - |

**Table 5.** Energy produced (GWh) by wind sources.

| YY/MM | Jan | Feb | Mar | Apr | May | Jun | Jul | Aug | Sep | Oct | Nov | Dec |
|---|---|---|---|---|---|---|---|---|---|---|---|---|
| 2012 | 135.589 | 113.968 | 107.132 | 119.239 | 85.230 | 50.047 | 87.253 | 85.406 | 67.077 | 105.527 | 109.182 | 144.190 |
| 2013 | 169.904 | 117.974 | 142.701 | 133.527 | 90.659 | 85.927 | 96.248 | 76.311 | 113.311 | 92.398 | 128.269 | 125.096 |
| 2014 | 108.989 | 104.726 | 146.570 | 95.313 | 79.403 | 103.633 | 78.927 | 76.347 | 115.160 | 136.171 | 110.114 | 174.390 |
| 2015 | 166.831 | 178.541 | 137.746 | 156.511 | 85.182 | 109.133 | 58.733 | 96.400 | 99.546 | 122.871 | 131.708 | 109.634 |
| 2016 | 168.025 | 158.043 | 117.710 | 100.034 | 92.290 | 86.808 | 58.202 | 130.623 | 62.081 | 117.247 | 137.765 | 197.868 |
| 2017 | 164.548 | 118.647 | 161.054 | 96.007 | 91.217 | 79.196 | 110.027 | 120.656 | 134.280 | 137.672 | 91.207 | 199.553 |
| 2018 | 134.301 | 154.501 | 116.361 | 104.380 | 113.180 | 68.789 | 60.628 | 90.956 | 103.588 | 133.082 | 118.070 | 120.288 |
| 2019 | 157.997 | 169.325 | 133.195 | 85.012 | 91.329 | 84.023 | 56.691 | 89.461 | 87.392 | 60.490 | 145.321 | 156.751 |
| 2020 | 167.681 | 176.257 | 188.932 | 127.131 | 124.034 | 68.726 | 71.232 | 78.550 | 106.877 | 92.471 | 112.558 | 162.680 |
| 2021 | 173.859 | 130.717 | 161.937 | 80.575 | 103.121 | 86.449 | 84.865 | 65.243 | 95.409 | 120.507 | 124.628 | 206.251 |
| 2022 | 218.051 | 143.500 | 202.822 | 138.760 | 72.054 | 84.036 | 74.342 | 92.768 | 102.289 | 91.577 | 126.057 | 140.510 |
| 2023 | 184.119 | 185.897 | 118.911 | 129.945 | 105.414 | - | - | - | - | - | - | - |

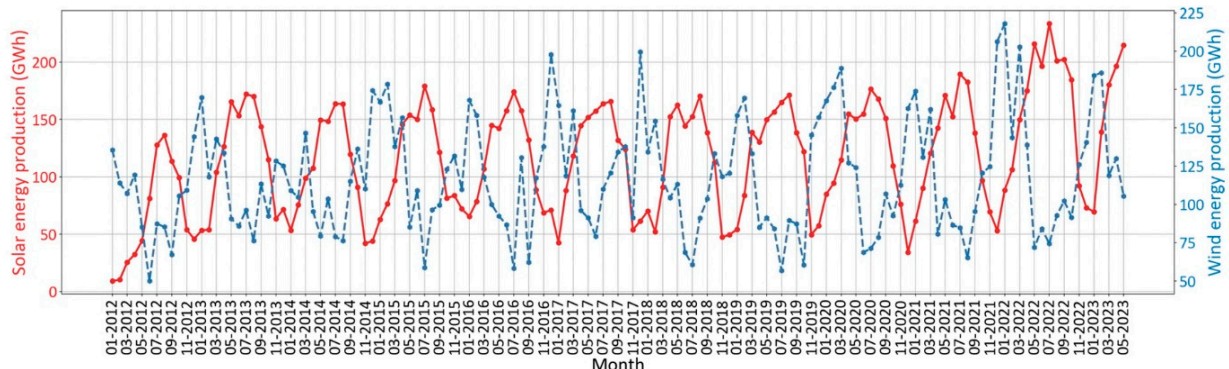

**Figure 11.** Plot of the time series: solar energy production in solid red line and wind energy production in dashed blue line. Source: Authors' work.

The electricity generated from renewable energy sources is dependent on the source, which, in the case of solar and wind power, is inherently intermittent. As a result, there is a need for the produced energy to be stored and used when required. Predicting expected energy production could help optimize decisions on the design of storage facilities and the purchase of storage equipment. Several methods can be used to forecast time series data. Autoregressive (AR) and moving average (MA) models, as well as combined ARMA models, are a common and classic approach, that is relatively inexpensive and quick to implement from a computing perspective. These models tend to give relatively good results when used for short-term forecasting of simple time series [24]. The Box–Jenkins method [25] is

often used to determine the appropriate order of such models. Furthermore, seasonality in the data can also be taken into account by extending the model to a seasonal autoregressive moving average (SARMA) model. Although a lack of suitable data prevented weather factors from being directly included as an exogenous variable in the modeling process, their impact on the primary predicted variable is observable and has an indirect impact on the modeling process. On this basis, a model of this type was selected for the purpose of this study. The approach and results are described in Section 4.

## 4. Forecasting Approach

### 4.1. Step 1: Data Preprocessing

#### 4.1.1. Splitting the Dataset

For the purpose of the study, both available datasets (solar energy and wind energy) were split into train data subsets from January 2012 to December 2022 and test data subsets from January 2023 to May 2023.

#### 4.1.2. Observation and Stationarity Check

As the measurements are taken on a monthly basis and energy production is dependent on weather conditions, we expect the data to show some seasonality. This is somewhat apparent in Figure 11, but we can also test this hypothesis through additional visual analysis by plotting the data in several specific ways, as shown in Figure 12.

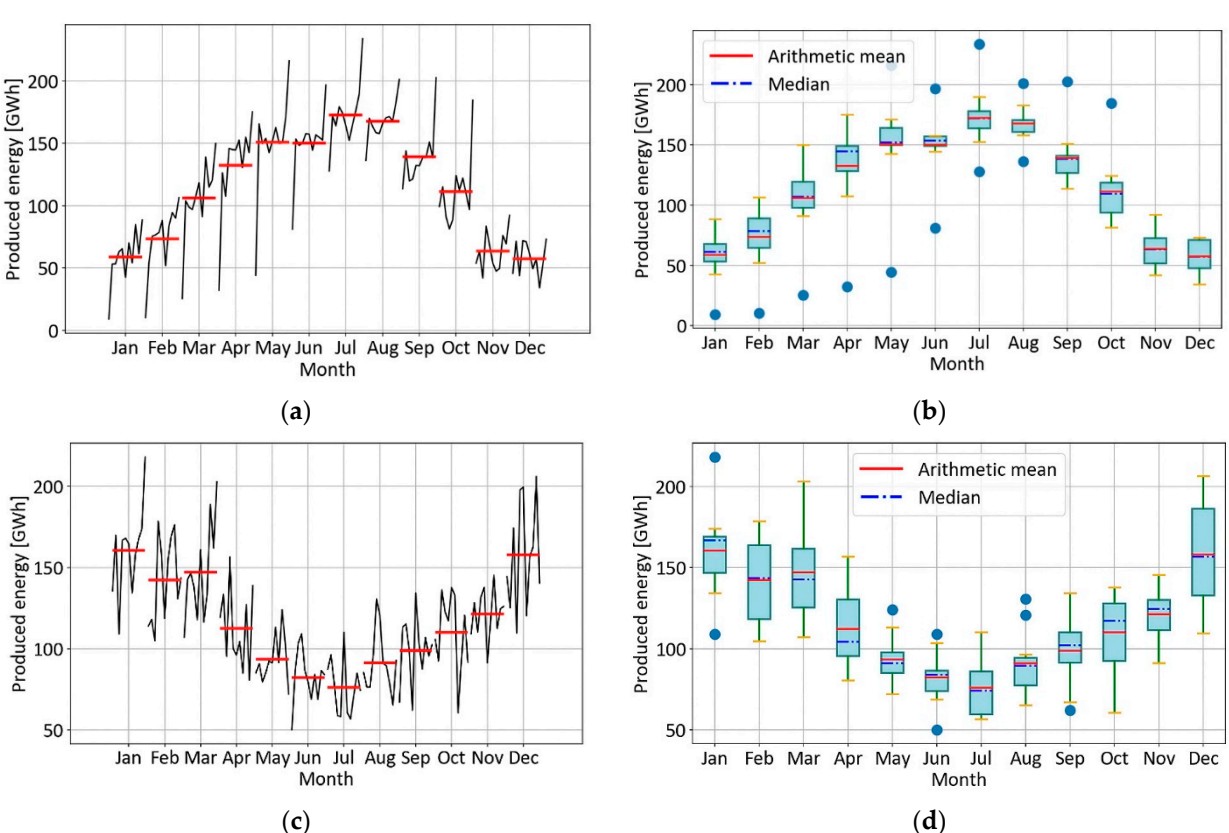

**Figure 12.** Seasonal subseries and Box plots: (**a**) Seasonal subseries plot of monthly solar data; (**b**) Box plot for solar energy data grouped by months; (**c**) Seasonal subseries plot of monthly wind data; (**d**) Box plot for wind energy data grouped by months. Source: Authors' work.

If we examine the graphics and the plots of the averaged values in Figure 12, we can see that despite the presence of several outliers, a seasonal pattern is emerging. For the solar data, there is a clear tendency for energy produced to increase in the summer months due to the characteristics of the summer season—longer days and a stronger sun.

Conversely, wind turbine energy production tends to decrease during the summer months, but increases during colder seasons such as spring, autumn and winter due to increased wind speeds. The oscillating and slowly decaying nature of the autocorrelation coefficients shown in Figure 13 further supports this observation. A high positive correlation can be observed at each 12th lag (i.e., lags 12, 24, 36, etc.), corresponding to similarities in energy production for each calendar month, from January to December, for each calendar year. Furthermore, there is also a high negative correlation at each half period (i.e., lags 6, 18, 30, etc.), which is consistent with data having a periodicity of 12.

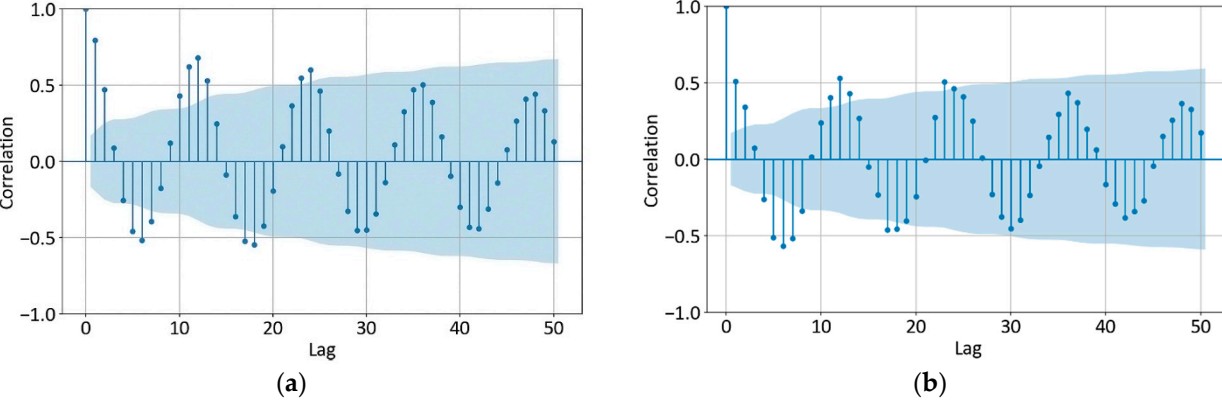

**Figure 13.** Autocorrelation function plots of (**a**) solar energy and (**b**) wind energy. The blue shaded areas represent the confidence intervals. Source: Authors' work.

### 4.1.3. First-Order Seasonal Differencing

ARMA models require the data to be stationary. By accepting the fact that the series are seasonal and this inherently implies their non-stationarity, we decided to perform seasonal differencing of the data as a next step before proceeding further. Seasonal differencing is a technique that can be used in time series analysis to remove the seasonal component of a time series. This is done by taking the difference between the current value and the corresponding value from the previous season [26–28]. This reduces our two sets of data by 12 observations, or one year each. However, data from distant periods are not expected to hold as much statistical significance as data from more current periods. It can be seen in Figure 14 that the seasonal nature of the data is largely removed because of the differencing that was done.

As seasonal differencing cannot guarantee the stationarity of the data, i.e., that the observed values are independent of time and there is no trend, we further test the seasonally differentiated datasets using the Augmented Dickey Fuller (ADF) [29] and Kwiatkowski–Phillips–Schmidt–Shin (KPSS) tests [30]. The results of the stationarity tests carried out for both datasets are shown in Table 6.

**Table 6.** Results from applying the ADF test and KPSS tests on the data and interpretation, after first-order seasonal differencing.

| Test | Parameters | | For Solar Seasonal Differentiated Data | For Wind Seasonal Differentiated Data |
|---|---|---|---|---|
| ADF | statistic | | −3.464 | −6.705 |
| | *p*-value | | 0.009 | $3.814^{-9}$ |
| | number of lags | | 11 | 11 |
| | number of observations | | 108 | 108 |
| | critical values | 1% | −3.492 | −3.492 |
| | | 5% | −2.889 | −2.889 |
| | | 10% | −2.581 | −2.581 |
| | is stationary | | true | true |

**Table 6.** *Cont.*

| Test | Parameters | | For Solar Seasonal Differentiated Data | For Wind Seasonal Differentiated Data |
|---|---|---|---|---|
| | statistic | | 0.224 | 0.070 |
| | *p*-value | | greater than 0.1 | greater than 0.1 |
| | number of lags | | 5 | 3 |
| KPSS | | 10% | 0.347 | 0.347 |
| | critical values | 5% | 0.463 | 0.463 |
| | | 2.5% | 0.574 | 0.574 |
| | | 1% | 0.739 | 0.739 |
| | is stationary | | true | true |

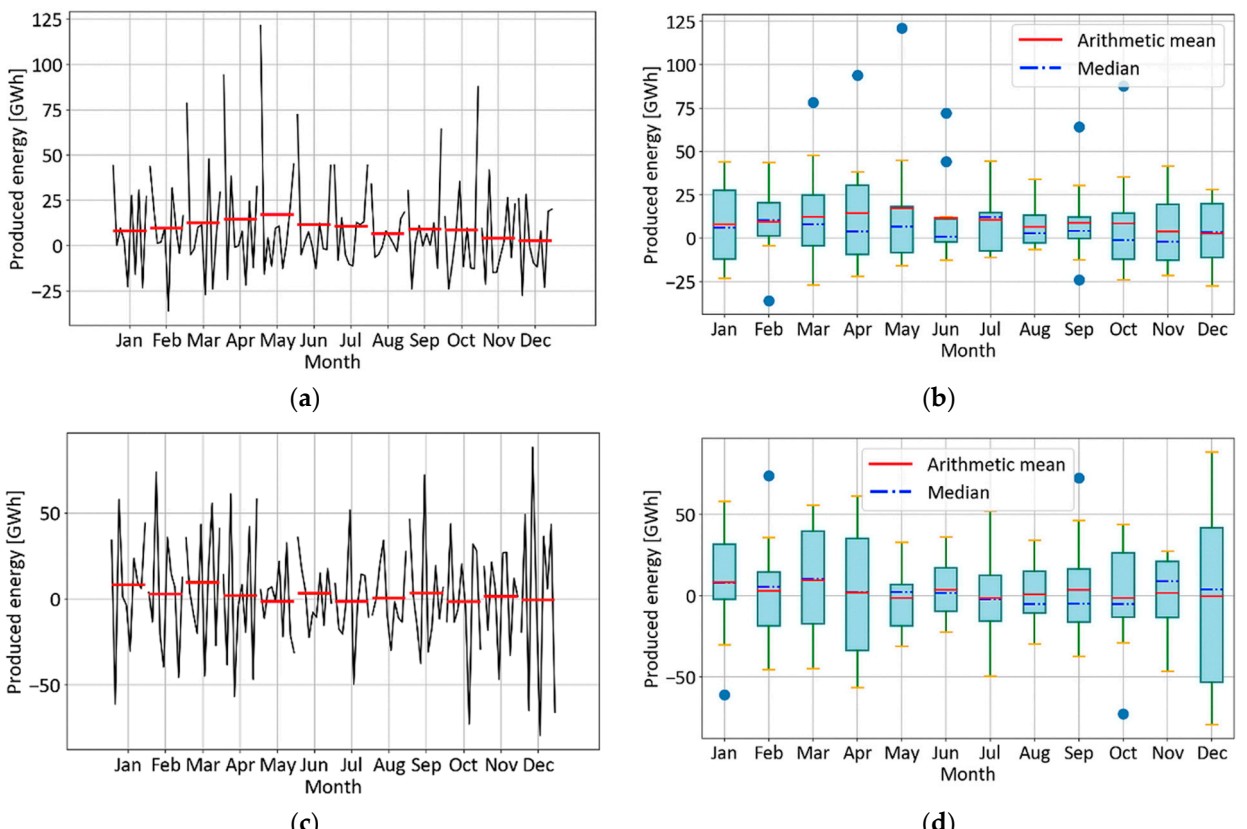

**Figure 14.** Seasonal subseries plots and Box plots after first seasonal order differencing: (**a**) Seasonal subseries plot of monthly solar data; (**b**) Box plot for solar energy data grouped by months; (**c**) Seasonal subseries plot of monthly wind data; (**d**) Box plot for wind energy data grouped by months. Source: Authors' work.

The conclusion from the tests is that both series are stationary after seasonal differencing. If either test fails, further transformation or differencing of the data must be applied to achieve stationarity.

### 4.2. Step 2: Estimating Model Parameters

Since neither the PACF nor the ACF, shown in Figure 15, have the characteristic cutoffs indicative of a pure AR or MA process [25], taking into account the test results in Table 6, seasonality and the need to differentiate the data, we presume that the seasonal autoregressive integrated moving average (SARIMA) model would be suitable for forecasting the generated electrical energy.

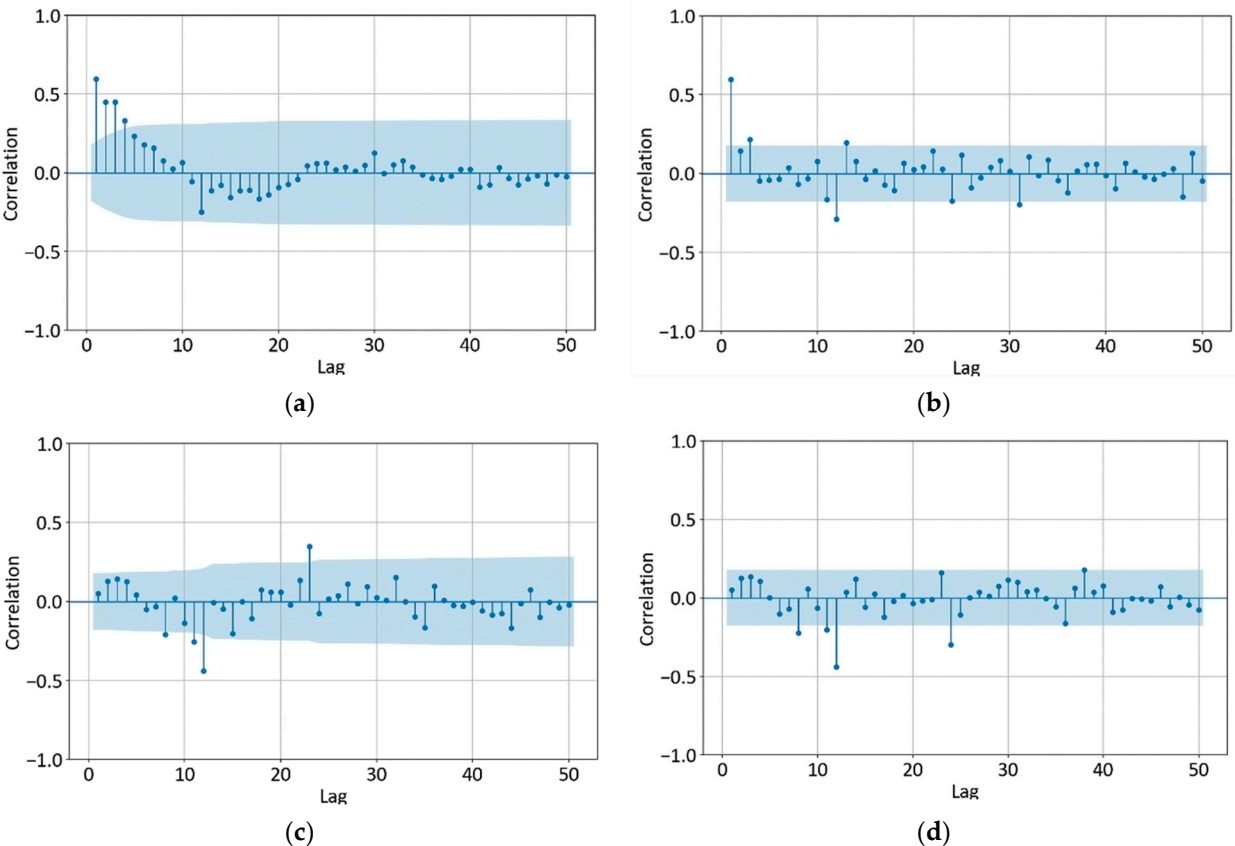

**Figure 15.** ACF and PACF plots after first seasonal order differencing: (**a**) ACF of solar data; (**b**) PACF of solar data; (**c**) ACF of wind data; (**d**) PACF of wind data. The blue shaded areas represent the confidence intervals. Source: Authors' work.

The values of the autocorrelation function (ACF) and partial autocorrelation function (PACF) coefficients of the seasonally differentiated data, shown in Figure 15a,b, suggest that suitable orders for the model of the solar data might be some of the combinations among the range of the following parameters: p = 1, 3; d = 0; q = 1, 2, 3, 4; P = 1, 2; D = 1; Q = 0; S = 12.

For the wind data model, based on Figure 15c,d, a possible order might be in the ranges: p = 0; d = 0; q = 0; P = 1, 2; D = 1; Q = 1; S = 12.

In order to select a relatively accurate and simplified model based on the ACF and PACF of both time series, we could choose model parameters corresponding to the lags having high correlation, but also those that occur at an earlier stage in the lag order.

In the PACF of the solar data, we observe the highest correlation above the confidence interval at lag 1, on the basis of which we define an AR ordering of *p* = 1.

After analyzing the lags 12, 24, 36, etc., which might indicate seasonal correlation, we observe a high value at lag 12; therefore, we choose P = 1 as seasonal order for AR. Similarly, based on the ACF plot, we choose q = 1 for the MA part of the model and Q = 0 for the seasonal part, as there are no lags with significant values indicating seasonal correlation. Thus, the model takes the form SARIMA(1,0,1)x(1,1,0,12).

In a similar way, we choose the order of the model describing the energy produced by wind turbines to be SARIMA(0,0,0)x(1,1,1,12).

The predictions that have been made by using the models are described above and shown in Figures 16 and 17:

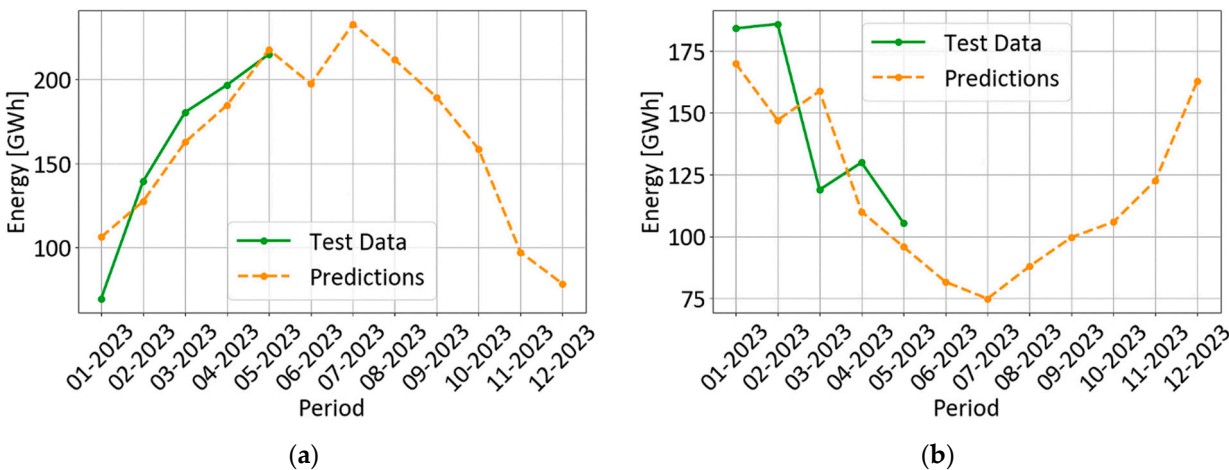

**Figure 16.** Monthly energy observations and twelve-month prediction: (**a**) Solar energy—model order SARIMA(1,0,1)x(1,1,0,12); (**b**) Wind energy—model order SARIMA(0,0,0)x(1,1,1,12). Source: Authors' work.

**Figure 17.** Estimated model twelve-month prediction: (**a**) Solar energy—model order SARIMA(1,0,1)x (1,1,0,12); (**b**) Wind energy—model order SARIMA(0,0,0)x(1,1,1,12). Source: Authors' work.

### 4.3. Step 3: Model Evaluation

An alternative approach to determine the appropriate model is to perform a grid search, i.e., generating and testing all combinations of parameters within certain bounds and evaluating the model results according to the given criteria—for example, Akaike information criterion (AIC) [31], Bayesian information criterion (BIC) [32], Hannan–Quinn information criterion (HQIC) [33] as well as the frequently-used metrics [34] as mean absolute percentage error (MAPE) and root mean square error (RMSE). The ranking of the best-performing models, estimated by a grid search within the following ranges—p = 0 ÷ 5; d = 0 ÷ 2; q = 0 ÷ 5; P = 0 ÷ 3; D = 0 ÷ 2; Q = 0 ÷ 3—is shown in Table 7 for the solar data and Table 8 for the wind data. The results of the manually determined models are also given for reference. The analysis suggests that there are models that produce slightly better metrics; however, the complexity of the processing that is required to estimate their parameters increases substantially.

**Table 7.** Grid search results for solar dataset.

| Model | Estimated | Lowest AIC, BIC, HQIC | Highest Log Likelihood |
|---|---|---|---|
| Order (p,d,q)x(P,D,Q,S) | (1,0,1)x(1,1,0,12) | (0,1,1)x(0,1,1,12) | (4,1,3)x(2,1,2,12) |
| Log Likelihood | −519.409 | −511.557 | −506.982 |
| AIC | 1046.818 | 1029.113 | 1037.965 |
| BIC | 1057.968 | 1037.450 | 1071.314 |
| HQIC | 1051.346 | 1032.499 | 1051.507 |
| Ljung-Box(L1)(Q) | 0.00 | 0.35 | 0.09 |
| Jarque-Bera(JB) | 0.80 | 2.10 | 1.74 |
| Prob(Q) | 0.97 | 0.56 | 0.76 |
| Prob(JB) | 0.67 | 0.35 | 0.42 |
| Heteroskedasticity(H) | 0.81 | 0.82 | 0.85 |
| Skew | 0.20 | 0.14 | 0.18 |
| Prob(H)(two-sided) | 0.50 | 0.54 | 0.61 |
| Kurtosis | 2.96 | 2.41 | 2.53 |
| MAPE | 15.71 | 16.71 | 19.14 |
| RMSE | 19.76 | 20.76 | 25.01 |

**Table 8.** Grid search results for wind dataset.

| Model | Estimated | Lowest AIC | Lowest BIC | Lowest HQIC | Highest Log Likelihood |
|---|---|---|---|---|---|
| Order (p,d,q)x(P,D,Q,S) | (0,0,0)x(1,1,1,12) | (2,0,2)x(0,1,1,12) | (0,0,0)x(0,1,1,12) | (0,1,1)x(0,1,1,12) | (3,1,4)x(2,1,2,12) |
| Log Likelihood | −561.708 | −556.138 | −561.773 | −559.480 | −550.758 |
| AIC | 1129.417 | 1124.275 | 1127.545 | 1124.960 | 1125.516 |
| BIC | 1137.779 | 1141.000 | 1133.120 | 1133.298 | 1158.866 |
| HQIC | 1132.813 | 1131.067 | 1129.809 | 1128.346 | 1139.058 |
| Ljung-Box(L1)(Q) | 0.06 | 0.43 | 0.02 | 0.05 | 0.38 |
| Jarque-Bera(JB) | 0.66 | 0.58 | 0.59 | 1.07 | 0.74 |
| Prob(Q) | 0.81 | 0.51 | 0.89 | 0.83 | 0.54 |
| Prob(JB) | 0.72 | 0.75 | 0.74 | 0.59 | 0.69 |
| Heteroskedasticity(H) | 1.18 | 1.07 | 1.17 | 1.26 | 0.94 |
| Skew | 0.11 | 0.05 | 0.11 | 0.18 | 0.04 |
| Prob(H)(two-sided) | 0.61 | 0.82 | 0.61 | 0.48 | 0.85 |
| Kurtosis | 2.72 | 2.67 | 2.74 | 2.72 | 2.62 |
| MAPE | 17.30 | 16.44 | 17.33 | 14.97 | 14.69 |
| RMSE | 27.52 | 25.03 | 27.53 | 25.74 | 24.64 |

Table 9 presents the predicted energy values for the period January–December 2023, estimated using the most efficient models, determined through grid search and the manually selected one.

**Table 9.** Twelve-month forecast by model.

| Months 2023 | Jan | Feb | Mar | Apr | May | Jun | Jul | Aug | Sep | Oct | Nov | Dec |
|---|---|---|---|---|---|---|---|---|---|---|---|---|
| **Model Order** | | | | | **Predicted Solar Energy, [GWh]** | | | | | | | |
| (1,0,1)x(1,1,0,12) | 106.09 | 127.32 | 162.55 | 184.53 | 217.65 | 197.22 | 232.89 | 211.64 | 189.20 | 158.43 | 97.11 | 78.33 |
| (0,1,1)x(0,1,1,12) | 106.04 | 124.06 | 159.58 | 186.06 | 208.15 | 199.99 | 226.19 | 215.15 | 192.30 | 164.14 | 107.32 | 93.56 |
| (3,1,4)x(1,1,1,12) | 109.06 | 123.94 | 152.37 | 186.88 | 213.48 | 196.62 | 226.61 | 221.44 | 193.04 | 164.61 | 113.67 | 97.55 |
| **Model Order** | | | | | **Predicted Wind Energy, [GWh]** | | | | | | | |
| (0,0,0)x(1,1,1,12) | 169.98 | 147.03 | 158.73 | 110.02 | 95.81 | 81.66 | 74.89 | 87.95 | 99.69 | 105.90 | 122.62 | 162.92 |
| (2,0,2)x(0,1,1,12) | 167.52 | 150.44 | 152.86 | 114.30 | 91.17 | 77.48 | 70.71 | 87.37 | 97.77 | 111.63 | 124.61 | 161.37 |
| (0,0,0)x(0,1,1,12) | 170.33 | 146.21 | 158.49 | 111.57 | 94.47 | 81.84 | 75.01 | 88.79 | 99.65 | 105.98 | 122.54 | 161.12 |
| (0,1,1)x(0,1,1,12) | 173.75 | 151.23 | 161.52 | 117.38 | 100.10 | 87.70 | 80.95 | 95.12 | 105.22 | 112.65 | 128.02 | 166.22 |
| (3,1,4)x(2,1,2,12) | 166.41 | 155.00 | 159.48 | 121.17 | 112.11 | 77.62 | 80.04 | 94.14 | 105.75 | 120.33 | 133.46 | 161.51 |

## 5. Simulation Environment and Source Code Repository

The simulations utilize the Anaconda scientific computing distribution (version 23.3.1) [35]. The code is written in Python programming language (version 3.10.9) [36]. To model time series, we employ the SARIMAX class from the Statsmodels library [37] for statistical and econometric analysis. Matplotlib [38] generates the plots in the Jupiter notebook.

## 6. Methods

A flowchart demonstrating how the forecasting approach was implemented is presented in Figure 18, following the steps:

Step 1.: Data preprocessing:

Step 1.1.: Split into train and test data subsets.

Step 1.2.: Observation and stationarity check:

- Generate seasonal subseries and Box plots of solar and wind data;
- Examine seasonal subseries and Box plots of solar and wind data for seasonal patterns;
- Generate ACF plots of solar energy and wind energy, using Statsmodels library implementation for time series plots;
- Examine ACF plots of solar and wind energy for oscillations that may indicate seasonal patterns.

Step 1.3.: Applying first-order seasonal differencing to eliminate seasonality (if detected):

- Examine seasonal subseries plots and box plots after first seasonal order differencing of solar and wind data to confirm lack of seasonality;
- Apply ADF and KPSS tests on the data after first-order seasonal differencing to verify stationarity in the transformed dataset.

Step 2.: Determining the appropriate model for the data:

- Based on the ACF and PACF plots, ADF and KPSS test results and indicated seasonality in the data, we choose SARIMA model for forecasting;
- Examine ACF and PACF plots of solar and wind data after first seasonal order differencing to determine model parameters—p, q, P, Q;
- Select model parameters d and D, based on ADF and KPSS test results, seasonal subseries and box plots;
- Use the SARIMAX model implementation of Statsmodels Python library for statistical and econometric analysis and generate a forecast based on selected parameters.

Step 3.: Model evaluation using criteria—AIC, BIC, HQIC and metrics—MAPE, RMSE. Compare the manually estimated model results with results obtained through grid search technique.

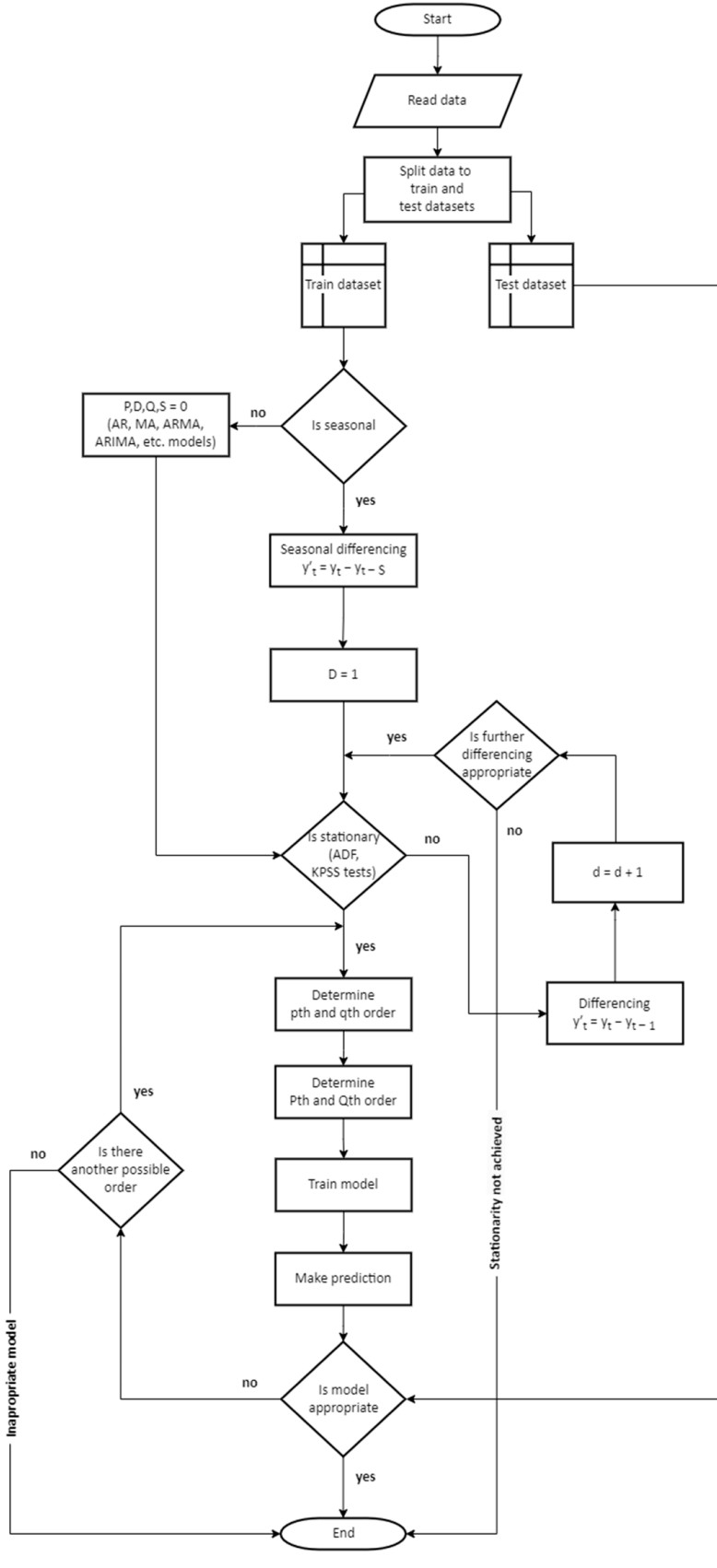

**Figure 18.** Flowchart of the forecasting approach implementation. Source: Authors' work.

## 7. Discussion

Questions about the use of RES are relevant for all countries across the world due to various circumstances. For industrialized countries that are dependent on the import of fuels and energy resources, energy security is of paramount importance. For industrialized countries that have plenty of energy resources, environmental security is the top priority. For developing countries, this is the fastest way to improve the social and living conditions of the population. The trends in the development of technical means and technologies for energy storage are logically related to the state of energy markets and the dynamics of energy mix formation.

Electric energy storage systems (EESS) are one of the key technologies and technical solutions for all branches of industry, economy and households, including energy transmission and distribution systems. EESS have the ability to address some critical characteristics of electricity, such as hourly fluctuations in demand and price. In the near future, EESS will become essential in emerging markets that are related to the use of more energy from RES in order to achieve reductions in $CO_2$ emissions and to introduce smart grids. EESS have three main roles:

- First, RES reduce the cost of electricity obtained off-peak when the electricity price is lower. It can be used during peak hours instead of purchasing electricity at higher prices.
- Second, in order to improve the reliability of electricity supply, RES systems support consumers when power grid failures occur, for example, due to natural disasters.
- Third, they maintain and improve the power quality, frequency and voltage. Regarding the needs of emerging markets, the grid is expected to solve problems associated with the use of large amounts of renewable energy (such as excessive power fluctuations and uncertainty). Instability and difficult predictability are two specific features of RES. Therefore, the growth of unstable generation volumes will increase the risk of losses and overloads if there is no energy storage.

To evaluate the efficiency of different types of generating capacity, the indicator "Levelized Cost of Electricity" (LCOE) was introduced. This indicator is the ratio of all life cycle losses to the amount of electrical energy produced during that life cycle. This includes capital costs, fuel costs, operational costs, maintenance costs, and many more. LCOE is the price at which the production of electrical energy from this or another energy source justifies the costs associated with this production. The lower the LCOE, the more profitable the investment in this energy source would be.

In September 2015, the investment bank Lazard announced a study according to which the LCOE of wind generation fell by 58% ($LCOE_{wind}$ = 37–81 USD/MWh) and that of solar generation by 78% ($LCOE_{solar}$ = 72–265 USD/MWh). The minimum and maximum values determine the degree to which this technology is used. Even then, wind energy turned out to be the most cost-effective energy source, delivering electricity at a price of 0.05 USD/kWh, compared to coal-fired power with average prices of 0.045–0.14 USD/kWh [39].

When assessing the impact of RES, special attention should be given to the so-called "grid parity" indicator. Grid parity is the distribution of generating capacities in which the LCOE of any generating source becomes equal to the price of energy for consumers. Different countries and different categories of users reach grid parity at different times. Considering that in 2019 the price of solar energy in China reached 0.04–0.11 USD/kWh [40], for most countries achieving grid parity will obviously come at the expense of their solar growth (including Bulgaria). However, the question of the potential of wind energy in our country remains unanswered. There are neither clear initiatives nor research in this direction nor storage facilities that were actually built. In fact, Bulgaria also has a hidden reserve for offshore wind power plants (Figure 19).

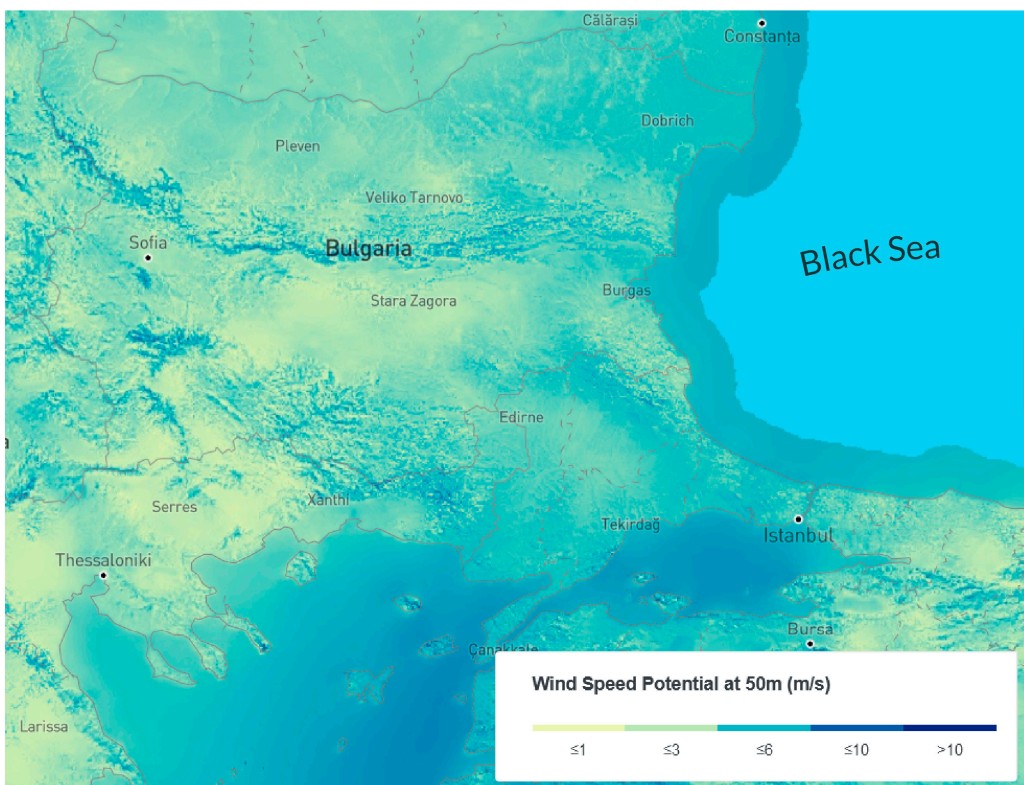

**Figure 19.** Wind speed potential at 50 m along the Black Sea coast [40].

Therefore, keeping in mind the benefit of the above-considered forecast models of solar and wind generation for Bulgaria for the last 11.5 years, here comes the question of what the most successful scenarios would be until 2030. Seasonality, cyclicality and the predominant share of solar energy are evident. That is why, in order to ensure a smooth and efficient energy transition, we are turning to long-term seasonal storage. According to [41], the required energy storage capacity needed to meet electricity demand for a year (given that the electricity source is solar, wind or a combination of both) is between 10 and 20% of the total annual consumption.

Two scenarios can be considered appropriate when it comes to estimating the percentage of storage capacity. Thus, the analyses carried out for RES generation in the country are applicable for the period from now until 2030. In addition to the useful conclusions, the data based on the following indicators is needed for an adequate assessment of storage capacity: A—Month of the year under consideration; B—Monthly electricity consumption, GWh; C—Monthly electricity consumption as a percentage of annual, %; D—Monthly electricity generation from wind energy, GWh; F—Monthly electricity generation from wind energy as a percentage of annual consumption, %; F—Monthly electricity generation from wind energy as a percentage of annual; G—Percentage of electricity storage required if the entire amount is produced from wind energy, %; H—Monthly electricity production from solar energy, GWh; I—Monthly electricity production from solar energy as a percentage of annual, %; J—Percentage of electricity storage required if the entire amount is produced from 50% wind energy and 50% solar energy, %; K—Percentage of electricity storage required if the entire amount is produced from solar energy, %.

In case solar power is the only source, we use the following equation to estimate the percentage of storage capacity:

$$K_{(n+1)} = K_{(n)} + I - C_{(n)}. \tag{1}$$

In case the solar and wind power have similar generation values (as may be the case for offshore wind development), another equation can be used:

$$J_{(n+1)} = J_{(n)} + 0.5 \times I_{(n)} + 0.5 \times F_{(n)} - C_{(n)}. \tag{2}$$

## 8. Conclusions

The energy transition scenario to be implemented in Bulgaria is clear: a short period of rapid growth with a certain share in the energy system and a search for the right mix of RES and storage systems, followed by moderate growth and gradual structural changes in the energy sector and infrastructure, but without any fundamental restructuring of the energy system. As RES provide a larger part of the total energy supply, the assessment of the required storage capacity is necessary for constructing an adequate renewable energy development scenario in order to achieve the goals for the first stage of the energy transition by 2030. A detailed optimization that takes into account the location and condition of transmission and distribution networks is necessary to obtain accurate estimates of optimal storage. Despite that fact, it is possible to use the current models of electricity generation from RES to estimate the storage capacity that is needed to balance the seasonal changes between the supply and the demand. An overproduction problem from RES can become a storage problem because excess production has a variable (often difficult to predict) value over time. For example, the overproduction of electricity from RES, as a source of cheap electricity, can lead to the production of hydrogen, which must be stored.

Based on the forecast, we may assume that if there is no significant increase in installed capacity for the future period and no serious impact from unforeseen external factors, the amount of energy produced from RES will likely remain steady or possibly increase.

The authors will deepen their research in the direction of integrating existing technological equipment into hybrid energy storage systems, e.g., hydrogen-batteries, ammonia-batteries, compressed air-batteries, etc., with a view to energy and resource efficiency in a period of energy transition.

**Author Contributions:** Conceptualization, D.K., R.K. and V.Z.; methodology, D.K. and R.K.; software, R.K.; validation, D.K. and R.K.; formal analysis, D.K., R.K. and V.Z.; investigation, D.K. and R.K.; resources, D.K. and R.K.; writing—original draft preparation, D.K. and R.K.; writing—review and editing, D.K. and V.Z.; visualization, D.K. and R.K.; contribution to the interpretation of the results, D.K. and R.K. All authors have read and agreed to the published version of the manuscript.

**Funding:** European Regional Development Fund within the OP "Science and Education for Smart Growth 2014–2020", Project CoC "Smart Mechatronic, Eco- and Energy Saving Systems and Technologies", № BG05M2OP001-1.002-0023.

**Data Availability Statement:** The code that was used to create the graphics and the model in this study is available from the GitHub repository at the following address: https://github.com/rkutkarska/energy-forecast (accessed on 31 August 2023).

**Conflicts of Interest:** The authors declare no conflict of interest.

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
