# Peer review of "High Penetration of Renewable Energy Sources and Power Market Formation for Countries in Energy Transition: Assessment via Price Analysis and Energy Forecasting"

_energies, doi:10.3390/en16237788_

Round 1
Reviewer 1 Report
Comments and Suggestions for Authors
The paper entitled “High penetration of renewable energy sources and power market formation for countries in energy transition: assessment via energy and price forecasting” aims at examining the power energy market formation with the increasingly large-scale and global penetration of renewable energy sources as primary energy sources. To do so the authors have used predictive models including Seasonal Autoregressive Integrated Moving Average (SARIMA). The paper is interesting and sheds some light on the examined case.
However, the following issues must be addressed before the paper can be published.
1) The paper’s literature is mainly based on the case examined. However, the authors must provide evidence why RES are important for the economy and how this importance has changed in the post COVID-19 era. Thus, a relevant section must be added in paper’s introduction (since there is not a literature review section). To do so, the following papers can be used:
- Ntanos, S., Skordoulis, M., Kyriakopoulos, G., Arabatzis, G., Chalikias, M., Galatsidas, S., Batzios, A. & Katsarou, A. (2018). Renewable energy and economic growth: Evidence from European countries. Sustainability, 10(8), 2626.
- Katsampoxakis, I., Christopoulos, A., Kalantonis, P., & Nastas, V. (2022). Crude oil price shocks and European stock markets during the Covid-19 period. Energies, 15(11), 4090.
- Drosos, D., Skordoulis, M., Arabatzis, G., Tsotsolas, N., & Galatsidas, S. (2019). Measuring industrial customer satisfaction: The case of the natural gas market in Greece. Sustainability, 11(7), 1905.
- Christopoulos, A. G., Kalantonis, P., Katsampoxakis, I., & Vergos, K. (2021). COVID-19 and the energy price volatility. Energies, 14(20), 6496.
- Cheng, Y. S., Chung, M. K., & Tsang, K. P. (2023). Electricity Market Reforms for Energy Transition: Lessons from China. Energies, 16(2), 905.
- EroÄŸlu, H. (2021). Effects of Covid-19 outbreak on environment and renewable energy sector. Environment, Development and Sustainability, 23(4), 4782-4790.
- Jin, S. (2020). COVID-19, climate change, and renewable energy research: we are all in this together, and the time to act is now. ACS energy letters, 5(5), 1709-1711.
- Liu, T., Nakajima, T., & Hamori, S. (2021). The impact of economic uncertainty caused by COVID-19 on renewable energy stocks. Empirical Economics, 1-21.
2) The paper lacks managerial and policy implication. Thus, a relevant section must be added after paper’s conclusions.
3) Moreover, there are no future research directions; a relevant section must be added as well.
4) The paper must be carefully proofread since there are several flaws in the use of English language.
Comments on the Quality of English LanguageThe paper must be carefully proofread since there are several flaws in the use of English language.
Author Response
Thank you very much for your useful comments, which have allowed us to improve the quality of the paper. We have done our best to address your comments and have uploaded a revised version of the paper which includes green (for methods, model and forecast procedure), yellow (for language and style) and blue (everything else) - coloured to indicate modifications to the original paper. Further details follow with your comments presented in italics and authors' responses are in normal text.
The authors would like to thank the reviewer for his precise and thoughtful comments and constructive criticism which has led to a better manuscript.
Reviewer 2 Report
Comments and Suggestions for Authors
The authors' article is devoted to an important and topical issue related to the study of the electricity market, including the use of renewable energy sources.
In most countries, renewable energy is the cheapest way to meet growing demand. In 2020, wind and solar developers won auctions with record low contract prices ranging from less than $20 per MWh to $50 per MWh. The offshore wind industry has seen an increase in scale in recent years (especially in Europe due to government support). This success should soon be replicated in emerging offshore wind markets in Asia and North America, with economies of scale leading to further cost reductions.
The growing share of renewable sources in individual countries' generation has opened new horizons to maximize investor interest in the sector and stimulate investment in battery technology. These changes are also associated with government policies that support the development of competition between traditional and alternative energy sources. Continued price declines alone will not protect renewable energy projects from a number of problems. The pace of economic recovery, increasing pressure on government budgets and the financial health of the energy sector as a whole are further exacerbating already existing energy policy uncertainties and financial constraints.
In the renewable electricity industry, there are two main categories of projects: those initiated by government actions (for example, through auctions, grants, subsidies) and market forces (for example, through corporate power purchase agreements, commercial projects). Each project category will face different opportunities and threats driven by two key variables: renewable energy cost trends; characteristics of national energy policy.
Based on the fairly high relevance and importance of the work performed, the results obtained will be useful for specialists in this field.
However, there are the following issues that should be clarified:
1. The introduction should strengthen the review of literature in the field of development of renewable energy, primarily solar and wind energy in various regions of the world. This will highlight the relevance and importance of the work carried out by the authors on analyzing the renewable energy market. In particular, the following works can be analyzed: https://doi.org/10.1016/j.egyr.2022.08.252; https://doi.org/10.1109/FarEastCon.2019.8934222; https://doi.org/10.1016/j.renene.2019.05.117.
2. A section “Materials and methods” should be added, where it is necessary to more clearly disclose the content of the methods used in conducting research.
3. It is necessary to explain how the predicted values of the parameters under consideration, presented in Figures 1-3, were obtained. Have machine learning methods based on artificial intelligence been used for forecasting purposes?
4. The work should present a structural diagram that includes the use of traditional energy sources and renewable energy, including possible hybrid options. This is necessary due to the fact that the use of various sources of renewable energy depends on a number of natural and man-made factors and therefore it is necessary to provide for various options for supplying energy to consumers, including the use of pumped storage sources, consumer-regulators of electricity, etc.
5. According to the data given in Table 4, a regression analysis should be carried out and mathematical models for calculating and forecasting electricity generation through the use of wind and solar energy should be presented.
6. It is necessary to dwell in more detail on the weather factors that were taken into account when analyzing electricity generation (Figure 12).
7. How were the correlation coefficients shown in Figure 13 obtained? A more detailed analysis of the obtained values of the correlation coefficients should be given.
8. Presented in section “4. Forecasting approach" numerous results should be presented in the form of a generalized research methodology that can be used by other scientists when carrying out similar work.
9. In Table 6, you should reduce the decimal precision (For Solar seasonal dif ferentiated data: -3.464075068003362), leaving the number: -3.464.
10. Figure 18 can be supplemented with similar data on wind speed potential in various regions of the world by conducting a corresponding analysis with the Black Sea coast.
11. Conclusions should be supplemented by numerous practical results given in the work. At the same time, prospects for further research in various regions of the world should be presented.
12. The list of references should include a larger number of works over the past 5 years on the topic under consideration.
Author Response

(The authors gave the same response as above.)

Reviewer 3 Report
Comments and Suggestions for Authors
The authors present some relevant data regarding RES deployment in Bulgaria and reason about the need for investment in energy storage to continue progressing in RES integration. A forecast is also made of solar and wind generation.
Nevertheless, the manuscript needs significant improvement for achieving the goals indicated in its title. The authors are asked to review the following issues:
- The title refers to "power market formation". No contents on this topic are apparently found in the manuscript. These words don't even appear in the text, except in the title. The same can be noticed about "price forecasting" announced in the title, but not done in the manuscript. Consider revising the title or adding these contents to the manuscript.
- The main findings should be indicated in the abstract.
- The source of figures is not indicated in all cases.
- A comprehensive state of the art is missing. No relevant academic studies concerning Bulgaria, forecasting or energy storage backup are cited.
- Line 194. Check if you mean "importer" or "exporter".
- Line 197 and others. Avoid the use of the sign "!" Reason what you want to express and justify it, avoiding subjective qualifications.
- Table 3. The first and the last rows are duplicated. The units for the last column are not indicated (%/MW/MWh?)
-Figure 6. Apparently the description of lines 211-218 do not correspond to Figure 6. Check.
- Line 235. Explain the acronym EES.
- Figure 7. Is the exchange presented representative, a sort of average value for the last years or is particular of the day and hour selected? The value would be greater selecting representative figures.
- Subfigure 8.c. The value of the figure is little without explaining the color code.
- Figure 10. The title refers to the last 24 hours, but of which date? Is it a representative date of a recent trend in prices?
- Figure 13. Explain what the shaded area is.
- Lines 355, 356. Explain the reason for that statement.
- Line 362, 363. Explain briefly what is seasonal differencing.
- Line 386. You say "on the basis of the tests". Explain that basis. The tests results have simply been collected at Table 6, without explanation.
- The discussion section does not discuss the results obtained in the previous section. Explain the value of the energy production forecast performed in the previous section and what can be concluded from these results.
- Which is the source for equations (1) and (2)? How have they been obtained? What is implied in applying them? Elaborate.
Comments on the Quality of English LanguageThe written English needs thorough revision.
Author Response

(The authors gave the same response as above.)

Round 2
Reviewer 1 Report
Comments and Suggestions for Authors
The authors have addressed all the amendments required and provided the needed revisions. Thus, the paper can be accepted for publication.
Comments on the Quality of English LanguageMost of the issues detected in English language are addressed. However, the manuscript must be proofread once again.
Author Response
Thank you very much for your useful comments, which have allowed us to improve the quality of the paper. We have done our best to address your comments and have uploaded a revised version of the paper, which includes pink-colored to indicate modifications to the original paper towards the language and style of the thesis. Blue-colored to indicate for added comments in the text additionally clarifying Figure 8c. Green-colored to indicate change in the abstract and clarification of the research method.
Further details follow with your comments presented in italics and authors' responses are in normal text.
The authors would like to thank the reviewer for his precise and thoughtful comments and constructive criticism which has led to a better manuscript.

Reviewer 2 Report
Comments and Suggestions for Authors
The authors have made the necessary changes. I recommend the article for publication.
Author Response

(The authors gave the same response as above.)

Reviewer 3 Report
Comments and Suggestions for Authors
The authors have not implemented all the asked changes. In addition, the pdf manuscript that the authors provide does not allow to see the color code changes that they mention at the cover letter.
- The main findings continue missing in the abstract. The abstract says what is done or plans to be done in the manuscript, but the main findings are still missing.
- Regarding the state of the art, doing a quick search using the manuscript keywords, some interesting titles emerge. Surely something useful could be found there. For instance, consider trying at Science direct with the keywords of the manuscript. (This reviewer has no interest in citing any particular reference, it is only a matter of showing that some apparently useful references appear dealing with renewable energy transition focused on Bulgaria, or considering the case of Bulgaria within the EU context).
- Comment 11: Subfigure 8.c. The value of the figure is little without explaining the color code.
Authors' answer: Thank you for pointing out this important issue. The authors totally agree with the reviewer. It was taken under special consideration and it was done.
Reviewer's reply: Apparently the change has not been applied in the modified manuscript. Could you specify where has it be done? Figures 8-10 are referred to in a sigle line of the text, without detailed explanation. If the necessary explanations can be skipped, it may indicate that the real value of those figures is little. Reconsider explaining the figures or deleting them.
Comments on the Quality of English Language
The quality of the written English has been improved.
Author Response

(The authors gave the same response as above.)
